# The Comparison of Retinal Microvascular Findings in Acute COVID-19 and 1-Year after Hospital Discharge Assessed with Multimodal Imaging—A Prospective Longitudinal Cohort Study

**DOI:** 10.3390/ijms24044032

**Published:** 2023-02-17

**Authors:** Kristina Jevnikar, Andrej Meglič, Luka Lapajne, Mateja Logar, Nataša Vidovič Valentinčič, Mojca Globočnik Petrovič, Polona Jaki Mekjavić

**Affiliations:** 1Department of Ophthalmology, University Medical Centre Ljubljana, 1000 Ljubljana, Slovenia; 2Department of Ophthalmology, Faculty of Medicine, University of Ljubljana, 1000 Ljubljana, Slovenia; 3Department of Infectious Diseases, University Medical Centre Ljubljana, 1000 Ljubljana, Slovenia; 4Department of Infectious Diseases, Faculty of Medicine, University of Ljubljana, 1000 Ljubljana, Slovenia

**Keywords:** COVID-19, SARS-CoV-2, retina, OCT, OCTA

## Abstract

This study aimed to quantify possible long-term impairment of the retinal microcirculation and microvasculature by reassessing a cohort of patients with acute COVID-19 without other known comorbidities one year after their discharge from the hospital. Thirty patients in the acute phase of COVID-19 without known systemic comorbidities were enrolled in this prospective longitudinal cohort study. Fundus photography, SS-OCT, and SS-OCTA using swept-source OCT (SS-OCT, Topcon DRI OCT Triton; Topcon Corp., Tokyo, Japan) were performed in the COVID-19 unit and 1-year after hospital discharge. The cohort’s median age was 60 years (range 28–65) and 18 (60%) were male. Mean vein diameter (MVD) significantly decreased over time, from 134.8 μm in the acute phase to 112.4 μm at a 1-year follow-up (*p* < 0.001). A significantly reduced retinal nerve fiber layer (RNFL) thickness was observed at follow-up in the inferior quadrant of the inner ring (mean diff. 0.80 95% CI 0.01–1.60, *p* = 0.047) and inferior (mean diff. 1.56 95% CI 0.50–2.61, *p* < 0.001), nasal (mean diff. 2.21 95% CI 1.16–3.27, *p* < 0.001), and superior (mean diff. 1.69 95% CI 0.63–2.74, *p* < 0.001) quadrants of the outer ring. There were no statistically significant differences between the groups regarding vessel density of the superior and deep capillary plexuses. The transient dilatation of the retinal vessels in the acute phase of COVID-19, as well as RNFL thickness changes, could become a biomarker of angiopathy in patients with severe COVID-19.

## 1. Introduction

Coronavirus disease 19 (COVID-19) is caused by severe acute respiratory syndrome coronavirus 2 (SARS-CoV-2), and even though several viral variants have emerged since the outbreak in December 2019, the proposed pathophysiological mechanism remains the same [1,2]. SARS-CoV-2 enters the cell by binding to angiotensin-converting enzyme 2 (ACE2), a key enzyme of the renin–angiotensin–aldosterone (RAAS) pathway and downregulates its activity [3]. Angiotensin-converting enzyme (ACE) and ACE2 are two of the key enzymes of the RAAS pathway. ACE catalyzes angiotensin-I (Ang-I) to angiotensin II (Ang-II), then is hydrolyzed to Ang 1–7 by ACE2. By binding to ACE2, SARS-CoV-2 downregulates its activity and creates an imbalance in the signaling effects of Angiotensin II (Ang II) and its receptor (angiotensin II type 1 receptor, AT1), resulting in the accumulation of Ang II. Increased serum Ang II leads to vasoconstriction, inflammation, cellular differentiation and growth, endothelial dysfunction, the formation of reactive oxidative species, and microvascular thrombosis. On the contrary, Ang II actions mediated through angiotensin II type 2 receptor (AT2) have a vasoprotective and anti-inflammatory role [4,5,6]. Nevertheless, AT2 has been proposed as an alternative entry point of SARS-CoV-2, blocking its possible protective role [7]. The resulting imbalance in the signaling effects of the RAAS pathway leads to endothelial dysfunction, which, combined with a hypercoagulable state, predisposes the patients to thromboembolic events [4,8]. ACE2 is expressed on the host cell surface of several tissues, including the retinal vascular endothelium, Müller glia and ganglion cells, and neurons in the inner nuclear layer [5]. In addition, another mechanism has been proposed, resulting in SARS-CoV-2-induced pyroptosis—an inflammatory type of programmed host cell death caused by direct infection of the retinal cells, leading to a production of cytokines, neuronal damage, and a hypercoagulable state [3,5,6,9,10,11,12,13]. The retina is especially susceptible to potential microvascular thrombosis, given its high metabolic demands and the fact that its plexuses contain terminal vessels without anastomotic connections [4,14,15]. COVID-19 retinopathy, encompassing several retinal findings, such as flame-shaped hemorrhages, cotton wool spots, dilated veins, and tortuous vessels, have been previously described. Furthermore, increased RNFL thickness and increased GCL thickness in several quadrants of the inner and outer EDTRS ring were described in the acute phase of severe COVID-19 [16]. The studies regarding the vessel density (VD) of the superficial and deep capillary plexuses have been inconclusive. While some found no differences regarding the vessel density [17] and the foveal avascular zone (FAZ) area [18], others reported decreased VD [18,19,20,21] and an increase in the FAZ area [20,22,23]. COVID-19 severity was found to affect the presence of retinopathy as a higher incidence of findings was reported in the moderate and severe course of the disease [4,17,24]. Several factors were shown to influence the severity of COVID-19, including age, male gender, and pre-existing comorbidities, especially cardiovascular disease, diabetes, and hypertension [25,26,27]. Underlying conditions enhance RAAS/Ag II imbalance; moreover, diabetes and resulting chronic hyperglycemia are known to compromise the innate immune system, which increases susceptibility to hyperinflammation and the development of the cytokine storm. This has been confirmed by increased inflammation-related biomarkers, such as C-reactive protein, serum ferritin, and IL-6 in diabetic patients with COVID-19 [8,25,28].

Furthermore, the polymorphisms in the genes of the RAAS pathway were shown to play a role in COVID-19 severity [29,30,31,32]. ACE insertion/deletion polymorphisms (rs4646994 and rs179752) were shown to affect the disease course. Individuals with a D/D genotype, which leads to an increased serum ACE concentration, were shown to have higher rates of pulmonary embolism and COVID-19-related mortality [18,25,26]. Conflicting results regarding the role of ACE2-related polymorphisms have been published [16,19,20,21]. It was hypothesized that the presence of A-allele, linked to increased gene expression of ACE2, resulting in higher ACE2 serum levels, and therefore and an increased number of viral binding sites, could increase COVID-19 severity [19,21]. Nevertheless, GG genotype and G allele carriers, associated with lower serum ACE levels, were shown to have an increased risk of SARS-CoV-2 infection and a more severe course of COVID-19 [33]. Therefore, the presence of A-allele was found to have a protective role, possibly explained by counterbalancing of the effects of increased Ang II resulting from the RAAS dysregulation [33]. Nonetheless, several studies found no association with disease susceptibility or severity [16,20,21]. An increased risk of severe COVID-19 was also associated with an AGTR2 polymorphism (rs1914711) [27]. It is noteworthy that ACE, ACE2, and AGTR2 polymorphisms have also been associated with hypertension, diabetes, coronary artery disease, and stroke [16,17,18,19,20,21], which complicates the risk estimation, as those comborbities alone are associated with an increased risk of COVID-19. Nevertheless, our previous study, focusing on patients without known comorbidities, showed an increased risk of COVID-19 retinopathy in males with the AGTR2-AA genotype of the rs1403543 polymorphism [34].

The longitudinal nature of our study is especially interesting given that up to 10% of patients are expected to develop long COVID-19, a multisystemic condition encompassing new onset thrombotic cardiovascular and cerebrovascular disease, dysautonomia, and Chronic Fatigue Syndrome [35]. Several mechanisms of long-COVID-19 pathogenesis have been proposed. It has been hypothesized that persisting reservoirs of SARS-CoV-2 in tissues could affect immune dysregulation and play a role in autoimmunity. In addition, long COVID-19 could result from the reactivation of herpesviruses, including the Epstein–Barr virus and human herpesvirus 6. Moreover, virus-induced dysfunctional signaling in the vagus nerve and the brainstem has also been postulated to lead to dysautonomia. In addition, endothelial dysfunction and microvascular thromboses are believed to have a role not only in the acute phase, but also long-term [35].

Optical coherence tomography (OCT) and optical coherence tomography angiography (OCTA) are non-invasive imaging modalities that provide high-resolution visualization of the retinal layers and the flow in its superficial and deep capillary plexuses, enabling the assessment of the microcirculatory changes in the retina [36]. This study aimed to quantify possible long-term impairment of the retinal microcirculation and microvasculature by reassessing the cohort of patients with acute COVID-19 without other known comorbidities one year after their discharge from the hospital. 

## 2. Results

The cohort consisted of 30 consecutive patients in the acute phase of COVID-19 without known systemic comorbidities who were reassessed a year after their discharge from the hospital. Baseline demographic and clinical characteristics are presented in Table 1. The cohort’s median age was 60 years (range 28–65) and 18 (60%) were male. Mean systolic and diastolic pressures were within normal limits. The most frequent symptom was fever (86.7%), followed by cough (76.7%), dyspnea (60%), chest pain (33.3%), headache (10%), anosmia (10%), and diarrhea (10%). The course was complicated by deep vein thrombosis and pulmonary embolism in one patient (3.3%). Patients were treated with dexamethasone, remdesivir, and oxygen, according to the guidelines. Twenty patients (66.7%) required oxygen supplementation. None of the patients presented with any signs or ocular symptoms, such as itching, photophobia, foreign body, conjunctivitis, or diminished visual acuity or exhibited them at 1-year follow-up. None of the patients in our cohort exhibited any of the long-COVID-19 symptoms. 

MVD significantly decreased over time, from 134.8 μm in the acute phase to 112.4 μm at 1-year follow-up (*p* < 0.001). MAD values were 96.1 μm and 93.1 μm, respectively, and even though the difference did not reach a statistical significance (*p* = 0.181), a tendency toward a decrease in the diameter can be observed (Figure 1). The comparison of the OCT parameters of patients in the acute phase of COVID-19 compared with 1-year follow-up is presented in Table 2, Figure 2. A significantly decreased RNFL was observed at follow-up in the inferior quadrant of the inner ring (mean diff. 0.80 95% CI 0.01–1.60, *p* = 0.047) and inferior (mean diff. 1.56 95% CI 0.50–2.61, *p* < 0.001), nasal (mean diff. 2.21 95% CI 1.16–3.27, *p* < 0.001), and superior (mean diff. 1.69 95% CI 0.63–2.74, *p* < 0.001) quadrants of the outer ring. There were no statistically significant differences between the groups regarding the vessel density of the SCP and DCP. Moreover, no significant differences were observed in the FAZ parameters, including FAZ area, FAZ perimeter, FAZ circularity, and axial ratio (Table 3 and Figure 3).

## 3. Discussion

In this study, we evaluated the evolution of retinal microvascular alterations assessed with multimodal imaging in patients without known ocular or systemic comorbidities in the acute phase of COVID-19 and at one-year follow-up. We found a significantly decreased MVD and a tendency toward a decrease in MAD at 1-year follow-up. These results are especially interesting, given that the significantly increased mean vessel diameters of both veins and arteries were present in hospitalized patients with severe COVID-19 in our previous study [16]. Similar results were reported, where a significantly decreased MVD and MAD were present after a 6-month follow-up; however, their baseline measurements were not performed in the acute phase of COVID-19 [37,38]. Our results can be explained by several mechanisms. First, the transient dilatation of the acute phase could reflect the vasogenic response to hypoxic and hypercapnic conditions of acute COVID-19 [15,37]. However, none of the confounding variables, including treatment with oxygen and inflammatory parameters, were shown to affect the vessel diameter in the acute phase [16]. Therefore, the hypothesis of SARS-CoV-2 induced RAAS dysregulation and the resulting endothelial dysfunction and injury seem more likely [9,39]. This is further supported by the findings from autopsy studies, where microvascular alterations such as small vessel thickening, vascular remodeling, and micro thrombosis were reported [40,41]. Hence, the longitudinal results and gradual decrease in the vessel diameters could reflect the regeneration of the vascular endothelium over time, a biphasic process where cellular proliferation and the resulting hypertrophy of the early phase are followed by normalization of cell density by pruning of excess cells in the circulation [42]. 

In addition, a significantly thinner RNFL in the outer ring of inferior, nasal, and superior quadrants was observed at follow-up. Given that the superficial layer of the retinal capillaries lies in the RNFL layer, this thinning possibly reflects the aforementioned decrease in the MVD. Similar results of thinner RNFL and GCL were reported over time [20,22,43,44,45]; however, vessel diameters were not assessed. The strength of our study is the simultaneous evaluation of those parameters, as the observed thinning seems to reflect regeneration rather than progressive SARS-CoV-2-induced RNFL atrophy. Moreover, no significant differences were found between the two groups in the VD of the SCP and DCP or any FAZ parameters, including the FAZ area, perimeter, circularity, and axial ratio. Notably, we also found no differences when comparing acute COVID-19 patients with a healthy control group [16]. While some of the previous studies are in line with our results [17,18], a decreased VD and an enlarged FAZ area were reported [18,19,20,21,46] and the observed differences were either stable [47] or more pronounced after a 3–8 month follow-up [21,21,22,23,23]. However, none of the studies reported any macroscopic changes on OCTA images, suggestive of either thromboembolisms or ischemia. It is noteworthy that only patients without known comorbidities that are known to affect the retina and retinal microvasculature, such as diabetes and hypertension, were included in our study. While we initially hypothesized that our results reflect the subtlety of acute changes, the long-term results may reflect the capability of the retinal intrinsic autoregulatory mechanisms to enable sufficient oxygenation to prevent ischemic damage even in the state of acute COVID-19 infection [14,15]. We acknowledge the limitations of our study. First, the baseline imaging was performed in the acute phase in the COVID-19 unit and was limited to patients admitted to the hospital during the study period. Moreover, only a proportion of initially enrolled patients were willing to attend the follow-up imaging. Second, OCT and OCTA have a limited reach; therefore, possible peripheral alterations could not be evaluated.

## 4. Materials and Methods

### 4.1. Study Design

A prospective longitudinal cohort study was conducted at the University Medical Center Ljubljana (UMCL) between December 2020 and May 2022. The study was approved by the Slovenian medical ethics committee (protocol ID number: 0120-553/2020/3) and adhered to the tenets of the Declaration of Helsinki. Written informed consent was obtained from all participants enrolled in the study. 

### 4.2. Patient Selection, Inclusion, and Exclusion Criteria

The patient cohort consisted of consecutive patients aged 18–65 with PCR-confirmed SARS-CoV-2 admitted to the COVID-19 unit of the department of infectious diseases UMCL. The exclusion criteria were as follows: systemic comorbidities (diabetes, arterial hypertension, hyperlipidemia, coronary artery disease, history of stroke), concomitant infectious diseases (HIV, HSV, VZV, CMV), systemic treatment linked to retinal toxicity, smoking, pre-existing ocular pathology (age-related macular degeneration and other retinal diseases, a history of glaucoma, high myopia (>−6)), and other conditions that could have affected the retinal morphology. The cohort was reassessed one year after their hospital discharge to confirm that they continued to meet the inclusion criteria. 

### 4.3. Study Protocol

The study was conducted in two locations; patients in the acute phase of COVID-19 underwent imaging in the COVID-19 unit of the Department of Infectious Diseases, UMCL, whereas the follow-up imaging of the same cohort a year later was performed at the Department of Ophthalmology, UMCL. All enrolled subjects were asked about the presence of the following ocular symptoms and signs: conjunctivitis, photophobia, itching, and diminished visual acuity. After dilating the pupils (1% tropicamide), fundus images, OCT, and OCTA were obtained using swept-source OCT (SS-OCT, Topcon DRI OCT Triton; Topcon Corp., Tokyo, Japan). The study protocol consisted of 4 images per eye: 2 color fundus images (one centered on the fovea, one on the optic disc), OCT centered on the fovea using the 7 × 7 mm scanning protocol, and OCTA centered on the fovea using the 3 × 3 mm scanning protocol. All the images were obtained by two doctors (KJ, LL). The appropriate full-body protective gown with the FPP 3 mask was worn in the COVID-19 unit. Patients’ electronic medical records were reviewed during hospitalization to collect the following demographic, clinical, and laboratory parameters: age, sex, presence of comorbidities (diabetes, arterial hypertension, hyperlipidemia, coronary artery disease, history of stroke), history of smoking, alcohol consumption, concomitant infectious diseases (HIV, HSV, VZV, CMV), time from the symptom’s onset or positive PCR to the day of fundus imaging, the presence of COVID-19-related symptoms, the need for oxygen, COVID-19-related treatment, and outcome. Laboratory parameters included lactate dehydrogenase (LDH), ferritin, CRP, procalcitonin, white blood cells, red cell distribution width (RDW), platelets, lymphocytes, D-dimer, and 25-OH-D3. 

### 4.4. Image Analysis

The eye with a better signal strength index was included in the analysis. Only images with a signal strength index above 60 were analyzed. All obtained imaging was independently reviewed by three researchers (KJ, AM, and PJM). Fundus photographs were screened for the presence of signs synonymous with COVID-19 retinopathy. Mean vein (MVD) and mean artery (MAD) diameters were assessed with the Automated Retinal Image Analyser (ARIA, V1-09-12-11) using a previously described method, in which the vessel diameters of the four main veins and four main arteries between 0.5 and 1 disc diameter from the optic disc margin were used to calculate the mean vein diameter (MVD) and mean artery diameter (MAD) [37]. OCT and OCTA images were automatically segmented by the built-in software (Topcon Corp., Tokyo, Japan), reviewed for the presence of abnormalities, checked for correct auto-segmentation, and manually readjusted if necessary. The thicknesses of the retinal nerve fiber layer (RNFL), ganglion cell layer (GCL), and retina in the four quadrants of the inner and the outer ring of the early treatment diabetic retinopathy (ETDRS) grid were exported using OCT Data Collector software (Topcon Inc., Tokyo, Japan). OCTA images of the superficial capillary plexus (SCP) and deep capillary plexus (DCP) were processed using MATLAB (MathWorks Inc., Natick, MA, USA), which was also used to analyze the foveal avascular zone (FAZ) [48]. The OCTA image was pre-processed using top-hat transformation to increase the vessel intensity. Extraction of vessel edges with a Canny edge detector was followed by morphological closure, image inversion, and removal of small elements to reduce the number of potential FAZ candidates. From the remaining candidates, the final FAZ was identified based on predefined area and eccentricity limits. Since the use of morphological operators reduces the FAZ segmentation precision, region growing was applied to the eroded FAZ. Pixels within 30% of the original region intensity were added. In cases where automatic segmentation did not produce satisfactory results, FAZ areas were outlined manually. FAZ area, perimeter, and circularity index were calculated. Once the center of FAZ was defined, ETDRS chart was superimposed to calculate the parafoveal vessel density (VD) in the four quadrants within the 3 mm circle of the center of FAZ. Each quadrant was separately binarized using the Otsu method. VD was expressed in percentage derived from the ratio of the total vessel area (white pixels) to the total area of the analyzed region (number of pixels in quadrant), a method previously described by Nicoló et al. [49]. The average vessel densities of the SCP and DCP were used for quantitative analysis. 

### 4.5. Statistical Analysis

Values are reported as mean (standard deviation, SD) or median (interquartile range, IQR) for numerical variables and as frequency (%) for descriptive variables. The normality was examined with the Shapiro–Wilk test. The differences in OCTA parameters before–after were evaluated as appropriate by dependent *t*-test or Wilcoxon rank-sum test. Linear mixed-effects regression was used to compare the OCT parameters between the groups. The subject was included as a random intercept to account for multiple measurements in each subject (N, S, T, I). P values of all pairwise comparisons were adjusted using the Benjamini–Hochberg method. Statistical analysis was performed with R statistical software (version 4.1.3, Vienna, Austria).

## 5. Conclusions

In conclusion, this is the first study to show the evolution of the retinal microvasculature and structure from the acute phase of COVID-19 to one-year follow-up. The transient dilatation of the retinal vessels in the acute phase of COVID-19 could become a biomarker of angiopathy in patients with severe COVID-19. Further longitudinal studies are warranted to assess possible long-term complications. 

## Figures and Tables

**Figure 1 ijms-24-04032-f001:**
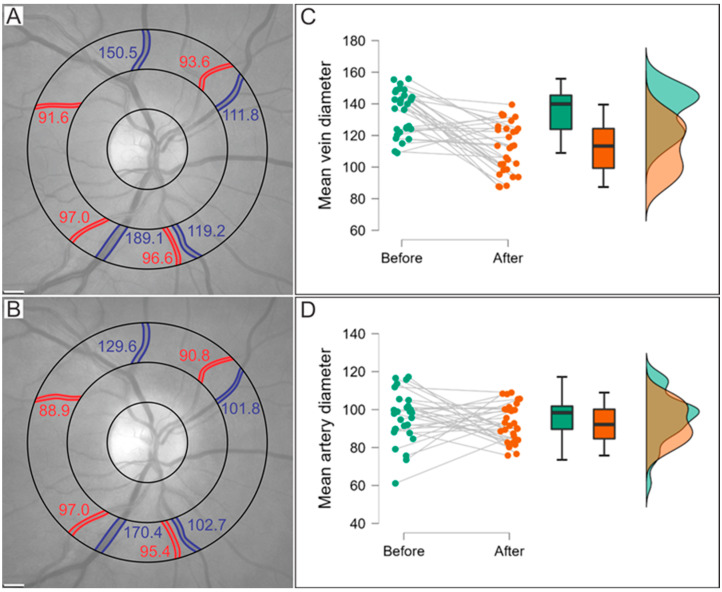
Comparison of the vessel diameters in the acute phase and at 1-year follow-up. Automatic retinal image analyzer software was used to analyze the mean the mean diameters (in μm) of the four main veins (blue) and four main arteries (red). There was a notable decrease in the vessel diameter from the acute phase (**A**) to 1-year follow-up (**B**). Raincloud plot of mean vein diameters (**C**); the mean difference (95% CI): 22.9 (17.0–28.9) and mean artery diameters (**D**); the mean difference (95% CI): 3.2 (−1.6–8.0).

**Figure 2 ijms-24-04032-f002:**
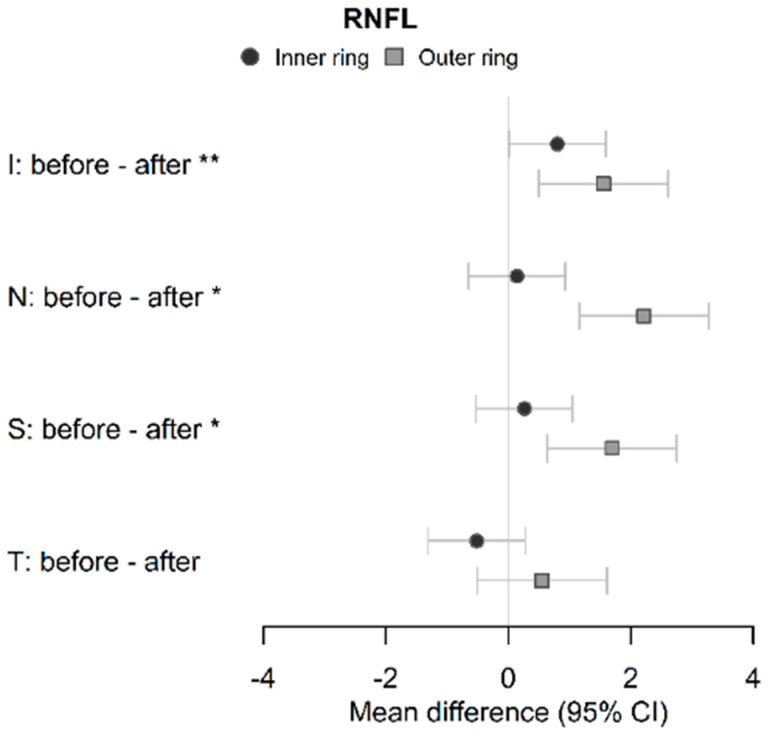
Forest plot of mean differences in RNFL thickness between the COVID-19 patients in the acute phase (before) and at 1-year follow-up (after) with 95% CI. ** *p* < 0.05 in the inner and outer ring; * *p* < 0.05 in the outer ring; RNFL, retinal nerve fiber layer; CI, confidence interval.

**Figure 3 ijms-24-04032-f003:**
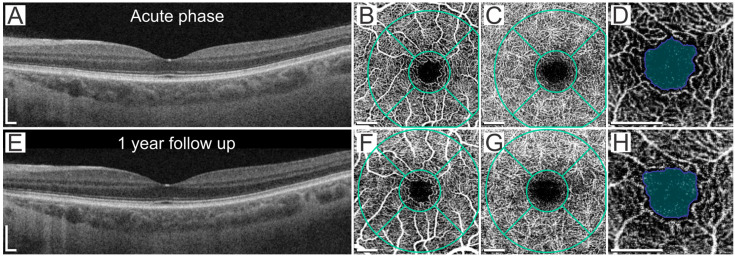
Comparison of the OCT and OCTA images of a single patient in the acute phase and at a 1-year follow-up. (**A**,**E**). OCT B-scan, note the thinner RNFL layer. (**B**–**D**,**F**–**H**). Segmentation of 3 × 3 mm OCTA images in the macular superficial capillary plexus (**B**,**F**) and in the macular deep capillary plexus (**C**,**G**). Foveal avascular zone area (**D**,**H**). Scale bar in all images: 0.5 mm.

**Table 1 ijms-24-04032-t001:** Demographic and clinical characteristics of patients in the acute phase of COVID-19.

	n = 30
Age, years (median, range)	60 (29–65)
Gender, male	18 (60%)
Systolic pressure (mean, SD)	132.2 (19.4)
Diastolic pressure (mean, SD)	77.7 (12.1)
COVID-19 clinical characteristics	
Duration of symptoms, days (median, IQR)	9 (5–11)
Fever	26 (86.7%)
Chest pain	10 (33.3%)
Cough	23 (76.7%)
Anosmia/ageusia	3 (10%)
Dyspnea	18 (60%)
Diarrhea	3 (10%)
Headache	3 (10%)
DVT/PE	1 (3.3%)
Laboratory parameters (median, IQR)	
LDH (μkat/L)	4.4 (3.5–5.2)
Ferritin (μg/L)	669 (310.5–1222.5)
CRP (mg/L)	33.5 (17–56.5)
Procalcitonin (μg/L)	0.04 (0.03–0.07)
White blood cells (×109/L)	7.5 (5.4–8.2)
RDW (%)	14.1 (13.4–14.6)
Platelets (×109/L)	222.5 (186.5–305.5)
Lymphocytes (×109/L)	1.2 (0.9–1.8)
D-dimer (μg/L)	705.5 (582.8–1013.3)
25-OH-D3 (nmol/L), n = 25	70 (48–84)
Treatment	
Dexamethasone	19 (63.3%)
Remdesivir	10 (33.3%)
Antibiotic	4 (13.3%)
Oxygen	20 (66.7%)

SD, standard deviation; IQR, interquartile range; DVT, deep vein thrombosis; PE, pulmonary embolism; LDH, lactate dehydrogenase; CRP, C reactive protein, RDW, red cell distribution width.

**Table 2 ijms-24-04032-t002:** Comparison of RNFL thickness between the acute phase and 1-year follow-up.

Circle/Side		RNFL		
		Acute Phase	1-Year Follow-Up	*p*-Value
Inner	I	29.18 (2.03)	28.38 (2.04) *	0.047
	N	24.42 (1.94)	24.28 (1.65)	0.656
	S	28.6 (2.14)	28.34 (1.93)	0.545
	T	19.18 (2.46)	19.7 (2.36)	0.207
Outer	I	41.07 (3.96)	39.51 (3.52) *	<0.001
	N	50.55 (5.92)	48.34 (5.68) *	<0.001
	S	40.95 (3.79)	39.26 (3.59) *	<0.001
	T	22.44 (2.74)	21.89 (2.2)	0.195

* *p* < 0.05; RNFL, retinal nerve fiber layer.

**Table 3 ijms-24-04032-t003:** Comparison of the OCTA parameters between the acute phase and 1-year follow-up.

	Acute Phase	1-Year Follow-Up	*p*-Value *
FAZ area (mm^2^)	0.21 (0.12–0.31)	0.23 (0.11–0.28)	0.584
FAZ perimeter (mm)	1.99 (1.41–2.26)	2.01 (1.40–2.18)	0.237
FAZ circularity	0.73 (0.65–0.79)	0.70 (0.64–0.75)	0.152
FAZ elipse	1.20 (1.12–1.28)	1.18 (1.15–1.36)	0.452
FAZ feret	0.63 (0.48–0.72)	0.63 (0.54–0.70)	0.271
FAZ convex area	0.23 (0.13–0.33)	0.25 (0.12–0.30)	0.344
Vessel density in SCP	35.65 (33.79–37.15)	35.89 (34.04–37.07)	0.777
Vessel density in DCP	40.59 (39.32–41.80)	40.90 (39.15–43.19)	0.700

* Dependent samples *t*-test or Wilcoxon rank-sum test; FAZ, foveal avascular zone; SCP, superior capillary plexus; DCP, deep capillary plexus.

## Data Availability

Data can be provided upon request.

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
