# Peer review of "The Comparison of Retinal Microvascular Findings in Acute COVID-19 and 1-Year after Hospital Discharge Assessed with Multimodal Imaging—A Prospective Longitudinal Cohort Study"

_ijms, 2023, doi:10.3390/ijms24044032_

Round 1
Reviewer 1 Report
I would like to thank the editors of the International Journal of Molecular Sciences for the opportunity provided for me to review the study entitled “The Comparison of Retinal Microvascular Findings in Acute COVID-19 And 1-Year After Hospital Discharge Assessed with Multimodal Imaging – a Prospective Longitudinal Cohort Study”. The authors investigated retinal microvascular in 30 patients with COVID-19 during the acute phase of the disease and one year after discharge. The investigators observed a significant decrease in mean vein diameter and retinal nerve fiber layer (RNFL) thickness. The authors concluded that the transient dilatation of the retinal vessels in the acute phase of COVID-19, well as RNFL thickness changes, could become a biomarker of angiopathy in patients with severe COVID-19. The study is well-written and addresses an important topic regarding the pathophysiology of COVID-19. Although the study lacks a control group, the findings of this study can provide additional insights into the pathophysiology of COVID-19 and pave the way for future studies regarding retinal angiopathy and microvascular involvement in COVID-19.
Author Response
We would like to thank the reviewer for their time and effort in reviewing our study. We really appreciate your thoughtful comments and thank you again for a favorable review.
Reviewer 2 Report
Jevnikar et al., in their paper have performed a prospective study on the effect of acute covid-19 infection on retinal microvasculature. While the virus mainly affects the respiratory system, several side effects of the virus that does not involve the respiratory system include myocardial infarction, acute ischemic stroke, and retinal microvascular abnormalities.
The paper is an extension of the published results where COVID-19 infection has been associated with alteration to retinal vasculature during the acute phase of the COVID-19 disease. While a 6-months follow up of retinal assessment is published, this paper provides a one year follow up and shows that the initial dilation of the retinal arteries and veins seen in the acute phase of infection is transient and is followed by its restoration to their normal diameter. This study thus shows that while changes in the retinal vasculature might be specifically occurring only in the acute phase of infection, a longer prospective study is required to show that there are no long-term effects of COVID-19 on retinal microvasculature.
Overall, it is a well written paper that acknowledges its limitations and also provides good explanation of the possible underlying mechanism that COVID-19 infection can have on retinal vasculature.
Author Response
We would like to thank the reviewer for their time and effort in reviewing our study. We really appreciate your thoughtful comments and thank you again for your favorable review.
Reviewer 3 Report
It's a very good study with novelty
Author Response
We would like to thank the reviewer for their time and effort in reviewing our study. The paper has been thoroughly revised and spelling checked using Grammarly software. Thank you for an overall positive review.
Reviewer 4 Report
This is a really interesting article concerning the OCT and OCTA evaluation of 30 patients that were affected by COVID-19, with a follow up of 1 year, trying to find potential differences in the retinal microvasculature.
The study is well organized, well-written and the study design is good. The Introduction and the Discussion section are concise enough and point out the key points of the study. The statistical analysis is well performed.
In my opinion, after a revision of the English, this manuscript could be accepted.
Author Response
We would like to thank the reviewer for their time and effort in reviewing our study. The paper has been thoroughly revised and spelling checked using Grammarly software. We really appreciate your thoughtful comments and thank you again for a favorable review.